# Informal Caregiving and Alzheimer’s Disease: The Psychological Effect

**DOI:** 10.3390/medicina59010048

**Published:** 2022-12-27

**Authors:** Emily Hellis, Elizabeta B. Mukaetova-Ladinska

**Affiliations:** 1School of Psychology and Visual Science, University of Leicester, Leicester LE1 7RH, UK; 2The Evington Centre, Leicester General Hospital, Gwendoline Road, Leicester LE5 4QF, UK

**Keywords:** Alzheimer’s disease, depression, anxiety, caregiver burden, traditional ideologies

## Abstract

*Background and Objectives*: People with Alzheimer’s disease and dementia in general benefit from home-based care as demonstrated via their better quality of life, increased lifespan, and delayed disease progression. Since currently nearly half of the dementia care is being provided by informal and unpaid caregiving, the health, wellbeing and quality of life of informal dementia caregivers is extremely important. *Materials and Methods*: We used a systematic review process with searches based upon the six elements from the “Quality of Life Scale for Informal Carers of Older Adults” with additional items on traditional and non-traditional caregiving ideologies, as well as caregivers’ experiences. *Results*: We identified 19 studies with primary data. Informal caregivers of older adults with Alzheimer’s Disease experience significant emotional strain, documented through increased levels of anxiety and depression, as well as increased caregiver burden and poorer quality of life, primarily due to caregiving ideologies, financial strain and a lack of support. *Conclusions*: Our findings suggest that caregiving should be a normative component of adult education to better prepare individuals with the mental and physical skills required for undertaking informal caregiving. They should also help inform policy makers to develop novel programs and services to both assist and reduce informal caregivers’ strain, whilst considering their different social and cultural contexts.

## 1. Introduction

Living in an ageing population has many benefits, both economically and socially, and yet it poses concerns for the healthcare system. In 2019, life expectancy was 79.4 years for males and 83.1 years for females. Due to the COVID-19 pandemic these estimates fell by 1.3 and 0.9 years, respectively [1]. Improvements in healthcare and the management of chronic conditions mean that people are living longer. However, as people age, they are often affected by one of more age-related diseases. This multi-morbidity means that older persons face a range of unique challenges leading to an increased need for care [2]. In addition to this, there is expected to be a 23.9% increase in people aged over 65 by 2039 [3]. With this comes the higher need for caregiving. It is, thus, appropriate to consider the impact caring for older persons has on the caregivers themselves and whether appropriate measures are in place to preserve their own health, wellbeing and quality of life (QoL).

Populations, projections and polling from Carers UK have estimated there to be ~9 million adults in the UK who are caregivers [4]. This does not take into consideration formal caregiving through the public and private sector. In 2016 the Office for Budget Responsibility investigated fiscal sustainability and public health spending, concluding that with the increasing health demands of our ageing population, the UK budget for healthcare would need a £13.3 billion increase within 5 years [5]. Demographic cost pressures in the years to come will push public spending ever upwards [6], thus finding a sustainable solution to the health and social care crisis remains a key challenge for generations to come. A greater reliance on informal caregiving may be considered as a potential source of relief for these underfunded systems. However, this increased demand for care must not become detrimental to informal caregiver’s wellbeing.

An informal caregiver may be identified as someone who provides some form of unpaid, ongoing assistance to a person with a chronic illness, age-related disease, or disability. This assistance is primarily with activities of daily living (ADLs): toileting, bathing, feeding, dressing and mobility assistance, and instrumental activities of daily living (IADls), such as financial assistance, transportation, shopping, cooking, cleaning and medication management [7]. Unsurprisingly, informal caregivers have been identified as key supportive figures in assisting older persons’ self-management of age-related diseases [8].

Currently, some of the most prevalent age-related diseases are the Dementias. This refers to a group of diseases which lead to progressive cognitive impairments and interfere with ADLs [9]. The most common type of Dementia is Alzheimer’s Disease (AD), a ‘neurodegenerative disease with insidious onset and progressive impairment of behavioural and cognitive functions including memory, comprehension, language, attention, reasoning and judgement’ [10], accompanied with disturbed perception and thought content, mood disorders and changes in behavior (i.e., aggression and wandering). Importantly, these behavioural and psychological symptoms are often associated with high levels of distress and anxiety for both the person with AD and their caregivers [11]. Therefore, it is not surprising that AD represents one of the main challenges for care providers of the elderly.

With the complexity of AD comes a high level of treatment and care which is extremely costly. Estimates per individual are set at around £32,350 per year with a total cost of £24.2 billion per year in the UK, £10.1 billion of which is attributable to informal caregiving and unpaid care [12]. With both medical professionals and scientists advocating for home-based care due to reported benefits for the individual with AD (increased lifespan and delayed disease progression [13]), the need to consider the informal caregiver’s QoL and wellbeing is extremely important, especially when one considers the breadth of research documenting links between informal caregiving and mental ill health [7,14], as well as the increased need for informal caregiving.

The Quality of Life Scale for Informal Carers of Older Adults was developed by Maltby et al. [15] based upon items from the Adult Carers Quality of Life Questionnaire [16]. The additional items developed in this scale considered that currents themes within general literature on caregiving of older adults came from traditional and non-traditional caregiving ideologies, as well as caregivers’ experiences. This was represented via six elements, five of which were based on Elwick’s questionnaire as seen in Figure 1 below.

The sixth new element ‘traditional caregiving role’ reflects a positive attribute which added to the caregivers’ QoL through feeling rewarded by their caregiving role and the relationship with those cared for. Thus, considering this additional element and the further five elements, it seems appropriate to consider these components as potential ‘risk factors’ affecting the QoL, health and wellbeing of informal caregivers of older individuals and those with AD. More specifically, due to their associations with mental ill health, these elements may be considered risk factors for Anxiety and Depression in these individuals [7,11,14].

In this review, we explore the association between Anxiety and Depression and the informal caregiving for people with AD as well as caregivers’ QoL, using the elements set out in the Quality of Life Scale for Informal Carers of Older Adults [15] as a basis. Findings from this study will inform future research within this area with the discussion of current and potential support for informal caregivers of individuals with AD.

## 2. Materials and Methods

A systematic study selection process was used to assess and interpret current research within this domain. During the planning stage search terms were determined based upon the Quality of Life Scale for Informal Carers of Older Adults [15] (Table 1). Systematic literature searches with no limit to study design and published until 01.10.2022 were carried out across several databases including Google Scholar, PubMed, PsycINFO and ResearchGate (Figure 2).

## 3. Results

Following on from Maltby et al.’s research [15], in order to better understand the psychological effects of informal caregiving for persons with AD, we explored further each element of the Quality of Life Scale for Carers of Older Adults. The results of these elements are summarised below (Table 2).

### 3.1. Traditional Caregiving Ideologies

A traditional caregiving ideology is one in which the needs of the person cared for are prioritised [17,18,19,20,21]. Caregiving is seen as an expectation, natural and virtuous, and often linked to high moral standards. A widely documented traditional ideology is Confucianism, a belief originating from Chinese culture which teaches individuals that they have a caregiving role within their family, with a focus on loyalty, interdependence and the maintenance of family harmony [22]. People are taught from a young age to respect their elders and that children are expected to care for older family members physically, financially and emotionally (filial responsibility) [17]. This has been found to hold true in more westernised society with later generation Chinese-American informal caregivers [17]. Due to this filial responsibility caregivers often have to make personal sacrifices to meet the individual caregiving needs of a person with AD. However, most caregivers report that they are willing to put the AD individuals’ needs above their own [23]. Furthermore, these caregivers often felt more positive and had better health due to the fact they were fulfilling their filial responsibility.

Lower levels of depression were found in informal caregivers with traditional ideologies, as well as great self-efficacy and the ability to respond more appropriately to some of the challenging behaviours common in AD [24]. Similar notions were previously reported with caregivers appearing to have felt psychological rewards through caregiving [25,26], by fulfilling filial responsibilities they found it easier to cope with stressors associated with informal caregiving of a person with AD. Informal caregivers who cared for a spouse with AD held traditional ideologies which came from their marital vows ‘in sickness and in health’ [27]. This was often associated with positive attitudes and lower levels of depression.

Opposing research has described a varying perspective, linking traditional caregiving ideologies to the informal caregivers feeling that they ‘have no choice’ in carrying out caregiving responsibilities [28]. Informal caregivers have reported making sacrifices in their personal and professional lives, such as missing social events and cutting down paid work [29]. These have all been found to be factors involved in worsening of their levels of anxiety and depression, as well as a decreased QoL [30].

### 3.2. Non-Traditional Caregiving Ideologies (Exhaustion Factors)

Non-traditional caregiving ideologies differ in that the informal caregiving is unexpected and often reflects a deviation from the caregiver’s life plan with no perceived reward [31]. Caregivers with non-traditional ideologies have reported feelings of having their lives temporarily stopped, they look at caregiving as an ‘obligation’ and mention ‘looking forward’ to when it was complete [27]. These non-traditional ideologies are often associated with higher caregiver burden, defined as ‘the level of multifaceted strain perceived by the caregiver from caring for a family member and/or loved one over time’ [30]. Caregiver burden was also associated with negative consequences, including a negative effect on the care provided, a decrease in QoL for the caregiver and the individual with AD, as well as deterioration in both physical and mental health. Higher levels of stress, anxiety and depression were also witnessed and have been directly related to the limited time informal caregivers give to themselves due to personal and professional sacrifices [32,33,34,35].

**Table 2 medicina-59-00048-t002:** Summary of included articles.

Authors	Method/Data Collection	Subjects	Country Setting	Findings
Miyawaki (2020) [17]	Structured interviews	*n* = 40 caregivers *Description*: 2nd, 2.5 and 3rd generation female Chinese-American caregivers caring after older relatives, some with dementia (NB. dementia type and number of carers for people with dementia not specified)	USA (Seattle and Houston)	Later generation caregivers had higher acculturation Filial responsibility remained high across generations Traditional caregiving was seen across all generations If the interviewed caregivers needed care in the future, their views upon this differed. Thus, caregivers from Seattle preferred the concept of longer-term care facilities whilst caregivers from Houston preferred being cared for by their children. This research emphasised the importance of caregiving attitudes and preferences being generationally and ethnically specific, and the importance of our understanding of this in a geographical context.
Sterritt and Pokorny (1998) [18]	Semi-structured interviews	*n* = 9 caregivers, with 3–8 years in caregiving; male and female African American Caregiver’s of relatives with Alzheimer’s Disease	South-Eastern USA	Found that caregiving is seen as a traditional family value Caregiving is thought of as an act of love Social support can be considered a mediator of caregiving burden Caregiving is considered to be a female role
Gray et al. (2009) [20]	Structured Interviews	*n* = 236 white, Hispanic, and Chinese-American women caring for relatives with either a diagnosis of Alzheimer’s Disease (or other dementia)	USA (San Francisco Bay area)	Attitudes and beliefs regarding AD/Dementia seen in Hispanic and Chinese caregivers may delay help-seeking activities for people with AD/Dementia. Hispanic and Chinese subjects were more likely to believe it to be a normal part of ageing diagnosable via a blood test than their white counterparts. This was attributed to their traditional and cultural beliefs.
Jones et al. (2011) [23]	Scale development	Questionnaires completed by 593 individuals. Filial concepts from scales using African-, Asian-, Euro-, Latino-, and Native American subjects were examined.	USA (Southern California and Native Americans)	Filial values predicted caregiving activities and caregiver health Three filial concepts were identified: Responsibility, Respect, and Care. These reflect attitudes and beliefs inherent in the complex multidimensional construct of filial values. A positive relationship between adult children professed filial values and their actual filial conduct was found. There was a stronger association between responsibility and care in males than females. Asians and African Americans displayed more filial responsibility.
Holland et al. (2010) [24]	Interventional study	*n* = 47 Chinese American dementia caregivers	USA (San Francisco Bay area)	Caregivers were found to report significant levels of distress, depressive symptoms, and also showed indications of resiliency—High levels of self-efficacy, positive caregiving experiences, and problem solving. Stronger beliefs in Asian values were associated with more normal cortisol patterns, less depressive symptoms, and greater self-efficacy, highlighting the salience of culture in shaping the caregiving experience of Chinese Americans.
Zhan (2004) [25]	Interviews	*n* = 4 Chinese-American caregivers of family members with AD	USA	There were ethnocultural and structural barriers facing the subjects; stigmatism of AD in the Chinese community, lack of knowledge about AD, a lack of culturally and linguistically appropriate AD services. There were negative impacts on mental and physical health.
Jones et al. (2001) [26]	Questionnaire based study	*n* = 50 Asian-American Women caregivers for aging parents (29 Chinese-American; 21 Filipino-American). All participants born outside of the USA.	USA	Involvement in caregiving was associated with health in Chinese-American women. Caregiving role integration was positively associated with all three perceived health measures in the Filipino group, but not in the Chinese group. Caregiving role satisfaction was consistently high in both groups. Caregiving role satisfaction and psychological well-being were significantly correlated for the combined group and for the Filipino caregivers. Total caregiving role stress was significantly correlated with overall health and current health only in the combined group. Support that helps to decrease role stress and to increase role satisfaction may be more effective than efforts to decrease the extent of role involvement.
Lawrence et al. (2008) [27]	In-depth interviews	*n* = 32 male and female caregivers of people with dementia (PwD)	UK (four socially and ethnically diverse south London boroughs: Lambeth, Southwark, Lewisham and Croydon)	Caregivers were identified as holding “traditional” or “non-traditional” caregiving ideologies. Within traditional ideologies caregiving was seen as a natural and honourable concept, something that is expected to happen. The majority of the South Asian, half of the Black Caribbean and a minority of the White British participants were found to possess a traditional ideology.
van de Ree et al. (2018) [29]	Structured Interviews	*n* = 123 informal caregivers of older adults (*n* = 22, 17.9%. had dementia; subtype not specified)	Netherlands (North Brabant)	Partners of the older adults provided more informal care than any other relative relationship. Female caregivers were 3-fold more likely to experiences relational problems due to caregiving. Majority of caregivers reported physical, mental and relational strain due to the intense nature of caregiving, particularly in the first six months.
Kang et al. (2016) [35]	Questionnaire based study	*n* = 87 caregivers of PwD (subtype unspecified)	Korea (Busan)	Caregiver burden, knowledge of dementia and levels of education predicted the quality of care given. Caregivers’ decreased QoLcame from caregiving burdens. Interventional and educational programmes aimed at reducing these burdens and increasing knowledge were deemed necessary to improve QoL and the quality of care given.
Shepherd-Banigan et al. (2020) [36]	Cross-sectional approach	*n* = 1509 familial caregivers of PwD within the Veteran Affairs (VA) programme (PwD = 44.9%)	USA (Nationwide)	Caregivers who care for veterans with trauma-based co-morbidities as well as cognitive decline reported high levels of depression, loneliness and financial strain even though they were part of the enhanced support system of the VA programme. Authors suggest a planned expansion of the programme to address these issues.
Harding et al. (2015) [37]	Secondary analysis	Data from 4 UK studies of informal caregivers of people with cancer (*n* = 105), dementia (*n* = 131; dementia subtype not specified) and acquired brain injury (*n* = 215)	UK (Sites not specified)	Caregivers’ burden was highest in those caring for acquired brain injury (ABI) and was followed by dementia caregivers’ burden. Total, subscale, and most individual elements of caregiver subjective burden differ between cancer, dementia, and ABI caregivers. However, concepts of duty, responsibility, and perception of financial situation were similar between the 3 groups. These should be considered when designing future intervention strategies to reduce caregivers’ burden in these groups.
Ku et al. (2019) [38]	Longitudinal study using interviews	*n* = 231 caregivers of PwD in a dementia clinic in Southern Taiwan	Taiwan (Tainan)	Behavioural disturbance [measured by the Neuropsychiatric Inventory (NPI)] showed no impact on the cost of care but was a significant predictor for caregiver burden. Caregiver burden was also associated with a functional decline in ADLs. Financial stability was associated with lower caregiver burden. These findings denote that financial assistance for low-income caregivers and educational training for behavioural disturbances are required to reduce caregiver burden.
Kang (2021) [39]	Secondary analysis	*n* = 956 unpaid family caregivers (National Long Term Care Survey, USA)	USA	The caregivers’ perceived burden was associated with financial strain, with variations due to familial relationships. The identification of these correlates can help with the development of effective interventions for caregivers’ burden.
Semiatin and O’Connor (2012) [40]	Interviews	*n* = 57 family caregivers of people with Alzheimer’s Disease	USA (Boston and Bedford)	Self-efficacy accounted for a significant percentage of the variance in positive aspects of caregiving after controlling for other factors commonly associated with positive aspects of caregiving including caregiver demographics, care recipient neuropsychiatric symptoms, and caregiver depression. High self-efficacy relates to caregivers’ perception of positive aspects of the caregiving experience.
Pendergrass et al. (2019) [41]	Cross-sectional study	*n* = 734 informal caregivers of PwD and other chronic illnesses	Germany (Bavaria)	There was an association between a higher experience of benefits, care duration, increase in depressive symptoms, increased physical grievances and a higher level of burden.
Horrell et al. (2015) [42]	Qualitative	*n* = 60 informal caregivers	New Zealand	The authors studied how emotions underpin informal caregiving. A caregiver’s choice of how they lived their lives was often influenced by their emotional attachment to the cared for, with higher attachment being associated with a decrease in wellbeing. The selflessness shown by the caregivers emphasised caregiving’s relational nature and challenged the prevalent perspective of caregiver burden documented previously.
Abreu et al. (2018) [43]	Cross-sectional study	*n* = 54 informal caregivers of PwD (*n* = 28 Alzheimer’s Disease, *n* = 12 vascular dementia, *n* = 9 mixed dementia, *n* = 2 Dementia with Lewy Bodies, *n* = 3 frontotemporal lobe dementia)	Portugal (Porto district)	Psychological distress was documented in half of the caregivers. Somatization, obsessive–compulsion, interpersonal sensitivity, anxiety, and paranoid ideation were seen in a large proportion of caregivers. The authors suggested placing focus on the alleviation of caregivers through education and additional support to help decrease their distress and burden
Laparidou et al. (2019) [44]	Qualitative	*n* = 35, 18 caregivers, 17 healthcare professionals	UK (Lincolnshire)	Primary stressors on caregivers came from lack of knowledge regarding Dementias and the challenge of diagnosis, often due to lack of understanding by healthcare-professionals. Secondary stressors were due the need for support and communication issues with healthcare professionals. The authors suggest that these stressors may be effecting the caregivers’ wellbeing r and may lead to an unnecessary move to institutionalised care for the care-recipient.

These non-traditional ideologies are reflected across families, with relatives of informal caregivers often refusing to provide support [27]. Feelings of guilt among relatives of individuals with AD and uncertainty were often aroused [30]. Underpinning these non-traditional ideologies is the sense that provision of care should not be down to the family/friends but to healthcare professionals, with informal caregivers often reporting immense pressure from family/friends to place the individual with AD into a residential care setting [45]. This often led to feelings of isolation and loneliness, effecting their mental wellbeing.

### 3.3. Financial Status

One must also consider the financial implications of caring for an older person, particularly the financial implications associated with AD, as mentioned previously. Many informal caregivers must forgo their full or part-time employment to dedicate their full time and energy towards caring for the older adult with AD. Full-time informal caregivers receive little to no support from the government—Currently, carer’s allowance stands at just £67.60 per week [46]. When one considers the financial savings mentioned earlier, with informal caregiving relieving the NHS of around £152 billion in care per year [47], it is devastating to think that the informal caregivers are provided with far less than a minimum wage job per week in order to provide this care, especially with the current cost of living crisis. Therefore, it comes as no surprise that the financial strains associated with informal caregiving have been linked to mental ill health and physical ill health within these informal caregivers [36].

Financial stress and mental ill health (i.e., increase in depressive symptoms and anxiety) are associated [48]. The experience of financial burden has been reported as five times greater when the caregiver has difficulties in balancing their caring role and their professional work [37,38]. The biggest financial strain is experienced among younger informal caregivers, who also have an increase in depressive and anxious symptoms compared to their older counterparts [39]. However, a tighter family bond was linked to both less financial strain and a decrease in depressive and anxious symptoms.

### 3.4. Personal Growth

Little research has commented on the positive effects of informal caregiving. It is, therefore, appropriate to consider the role of personal growth in informal caregiving as per the Quality of Life Scale for Informal Carers of Older Adults [15] in order to assess the need for future research focusing on this concept. From the limited research focused on positive effects of caregiving and its effect on personal growth, it appears that these positive caregiving experiences may act as a buffer for the effect of physical demands and psychological distress that informal caregiving has on a caregiver [40]. As well as this, the sense of personal growth, that comes from the positive experience of caregiving, has provided them with the ability to view their role as a caregiver with a more balanced perspective, leading to fewer reports of anxious and depressive symptoms [49]. In contrast to this, statistically significant correlations between depressive symptoms and a sense of greater benefits and personal growth from caregiving, a seemingly counterintuitive notion [41], have been found. However, this research concluded that personal growth is still able to occur from informal caregiving whilst experiencing depressive symptoms due to the demands of caregiving and the decline in health of a relative, spouse or a friend.

### 3.5. Ability to Care and Level of Support

An individual’s ability to care and the levels of support they receive from relatives and friends are appropriate to consider together. These themes are interchangeable, as documented by the findings that one’s ability to care is very much dependent on the level of support one is receiving [42,50,51]. As such, both one’s ability to care and the level of support one receives have both been associated with mental well-being within informal caregiving [43,44].

Informal caregiver’s confidence in themselves and their ability to care have a significant negative correlation with reported stress and poorer mental wellbeing [52]. This highlights the importance of considering support needs for informal caregivers in order to prevent additional health problems and prevent the practice of informal caregiving from occurring. Thus, it is not surprising that both anxiety and depression are reported to be common in informal caregivers of older adults with chronic care needs, i.e., cancer [32,34] or dementia [40,47,49], and seems to be closely linked to the level of support received and ability to care, much like that seen in individuals with non-traditional caregiving ideologies whose families did not offer support.

## 4. Discussion

The purpose of this research was to explore the six elements set out by the Quality of Life Scale for Informal Carers of Older Adults and their association with poor psychological wellbeing in informal caregivers of Older Adults with AD. As shown by the results, poor financial status, non-traditional caregiving ideologies and lack of support have been linked to higher levels or anxiety and depression in informal caregivers. This was also seen across some research relating to traditional caregiving ideologies; however, these were also seen as a protective factor towards mental ill-health, similar to that seen for personal growth [22,28]. Although it appears that these elements are related to anxiety and depression in informal caregivers of those with AD, this requires further research to establish the true relationship between these concepts.

### 4.1. Traditional Caregiving Ideologies

The lack of research surrounding high levels anxiety in informal caregivers of those with AD who hold traditional caregiving ideologies may be due to the positive outlook associated with traditional caregiving ideologies and fulfilling filial responsibilities [53]. As a result, traditional caregiving ideologies should perhaps be viewed as a protective factor for anxiety in informal caregivers of persons with AD as opposed to a risk factor.

Interestingly, however, previous research did find an association between higher levels of depression and traditional caregiving ideologies, suggesting that the significant burden, stress and time associated with providing informal care, particularly to those with AD, leads to increased level of depression [30]. A possible explanation for this is the individual feeling of being ‘trapped’ by the traditional ideologies caregivers have been brought up with. Additionally, caregivers may perceive that it is their duty and responsibility to provide this care, particularly if looking after a parent, as they feel they must care for their parent as their parent had once cared for them. With this comes a cost to their own health and wellbeing.

When searching for previous literature surrounding traditional caregiving ideologies and anxiety/depression, there was little discussion about support available for informal caregivers. Nevertheless, the informal caregiver’s traditional ideologies in respect to caregiving will need to be considered when conceptualizing ways in which novel programs and services can be developed to assist informal caregivers. This is important since due to informal caregivers’ traditional ideologies, they may be less likely to accept support from outside their family network, as they believe it is their filial responsibility to provide care. In addition to this, they may be less likely to seek professional help when experiencing depressive symptoms, as they may feel guilty that they are feeling emotionally strained from their informal caregiving, something which is expected and required of them by their families.

In conjunction with previous research, these findings provide guidance for future research in both quantitative and qualitative manner. Firstly, it would be of interest to use the Quality of Life Scale for Informal Carers of Older Adults [15] using subjects who are informal caregivers and measure levels of anxiety and depression using a tool such as the Hospital Anxiety and Depression Scale (HADs) [54] to determine an association between traditional caregiving ideologies and anxiety/depression levels. Secondly, it would be important to investigate how traditional views vary across different cultures and ethnicities, and whether it is these variations in traditional caregiving ideologies and teachings that cause the documented differences in psychological wellbeing in terms of anxiety and depression.

### 4.2. Non-Traditional Caregiving Ideologies (Exhaustion Factors)

In the Quality of Life Scale for Informal Carers of Older Adults [15] the non-traditional caregiving ideologies are measured as part of the ‘exhaustion’ variable. This variable is thought to encompass non-traditional ideologies in that an individual’s fears about the informal caring role and deviation from their life expectations loads on exhaustion factors. As seen in previous studies, non-traditional caregiving ideologies are based upon a deviation from one’s life plan and are often associated with caregiver burden and exhaustion [27,28,29,30]. It is, thus, not surprising that there is an association between holding non-traditional caregiving ideologies and an increase in caregiver burden, increased levels of anxiety and depression, feelings of isolation and guilt. All of these are contributing to a decreased QoL for both the informal caregiver and, as a result, the person with AD that is being cared for.

When looking at the statements in Maltby et al.’s (2020) questionnaire which measures these exhaustion factors/non-traditional caregiving ideologies [15], it is clear to see why previous research has documented a link between this element and anxiety and depression in informal caregivers. Some of the statements include ‘I am mentally exhausted by caring’ and ‘I feel I have less choice about my future due to caring’, both of which can easily be related to feelings of anxiety and depression. It seems appropriate to consider what assistance can be put in place to enable informal caregivers to provide the care needed whilst not deviating too far from their life plan, as well as what support they require to help relieve feelings of stress, anxiety and depression.

### 4.3. Financial Status

Understandably finance underpins informal caregiving, from the amount of money it saves the NHS each year, to the amount it costs informal caregivers themselves, both in giving up professional employment and the costs associated with caring for a person with AD. Financial status was found to be a constant throughout, in terms of being a factor associated with poorer mental wellbeing and QoL [37,47]. In particular, younger informal caregivers were often the ones that reported the higher levels of anxiety and depression, but this greater prevalence was reflected across various age groups of informal caregivers of those with AD, suggesting that financial hardship should be considered as a risk factor for anxiety and depression in the informal caregiving population [47]. Since the experience of financial strains and financial burden has been associated with difficulties balancing a formal caring role, measures need to be put in place to help support informal caregivers.

Since the combination of financial strain and poorer mental wellbeing are leading to a decreased QoL for these informal caregivers, firstly it seems appropriate to tackle the concept of financial aid. With caregiver’s allowance standing at £67.60 per week [36], and many informal carers forced to reduce hours or quite paid employment, more needs to be done to financially enable this caring to take place, especially when one considers the amount of savings informal care provides our public healthcare service. Secondly, with the increased financial strain among younger caregivers and resulting increased levels of anxiety and depression, it seems appropriate to consider the development of educational programs around financial management, as well as aid in finding employment with more flexible working hours. In addition to this, it would be appropriate to educate companies on the difficulties associated with informal care, which may lead to changing policy to better accommodate informal caregivers in the working environment.

### 4.4. Personal Growth

In informal caregivers of those with AD, personal growth appears to have a positive impact on anxiety and depression levels. Although research is limited, this is an extremely positive concept for informal caregiving. However, with some research indicating that depressive symptoms may still occur in line with feelings of personal growth in caregiving it is important to consider this further. For example, it is of psychological interest to further investigate specific caregiving experiences that are related to personal growth and a sense of achievement (for example, the impact of respite care where a volunteer or formal caregiver is assigned for a limited period of time to allow the informal caregiver time away from caring). By investigating this further, we will be able to inform policy ideas and help to facilitate more rewarding caregiving experiences for the informal caregiver and for the older adult being cared for. With this we will hope to increase the QoL of both the informal caregiver and the individual being cared for. In addition to this, the development of novel support programs and therapeutic interventions, which aim to educate and aid individuals with these more positive experiences of caregiving to help with personal growth, will be an appropriate support tool. Informal caregiving should not come at a cost to the physical and mental health of the caregiver, or their QoL, and these programs will enable a better QoL for informal caregivers.

### 4.5. Ability to Care and Level of Support

Informal caregivers have higher level of depressive symptoms, and they are associated with a lack of support and subsequent ability to care [42,50,51]. This lack of support was often reported to be from relatives and has been shown to tie into caregiving ideologies, with those holding non-traditional ideologies providing the least support and often leading to feelings of guilt, anxiety and isolation, whilst families holding traditional ideologies were reported to offer the most support [51]. In addition to this, those brought up with traditional ideologies reported feeling more prepared for informal caregiving and as such showed a better ability to care and hence lower depressive symptoms [44].

With previous literature suggesting a link between lack of support and feelings of ability to care, anxiety and depression in informal caregivers [55], it is important to consider the next steps for this premise. We suggest that efforts should be made to make clear distinctions between the factors affecting the QoL of the informal caregiver as this may lead to different policy responses. For example, it may be more appropriate to provide respite for informal caregivers rather than looking at ways in which they can continuously perform their caregiving obligations. In line with this, the potential of the use of technology is a concept to consider. For instance, they can help aid the 24/7 caregiver hotlines to provide support when traditional resources are unavailable. Dementia patients often have disturbed sleep, which causes the caregiver to also be up at odd hours. Telephone, computer, or video supports can help caregivers through these difficult times. Substantial progress has been made recently to aid both formal and informal caregiving [56,57,58,59,60], thus this may be an avenue to contemplate with future research in combination with findings from this study. However, further research is needed on caregivers’ views, with a solutions-based approach that will identify caregivers’ problems and at the same time will provide possible solutions to address these based on the perceived needs of the caregiver.

## 5. Conclusions

Overall findings highlight a link between financial strain, anxiety and exhaustion for informal caregivers of older adults with AD. In addition, higher levels of depression are associated with financial strain, exhaustion and traditional caregiving ideologies. Non-traditional caregiving views feed into the concept of exhaustion and were measured as such in the Quality of Life Scale for Informal Carers of Older Adults. These findings suggest that caregiving should be a normative component of adult education, in order to better prepare individuals with the mental and physical skills required for undertaking informal caregiving. These findings will help inform policy makers to develop novel programmes and services to both assist and reduce informal caregivers’ strain, taking into account their different social and cultural contexts. 

Most of the studies included in the current study focused on Asian or Asian-American caregivers, arguing for the need for more studies to address broader range of cultural approaches to caregiving. We feel both a quantitative approach using the Quality of Life Scale for Informal Carers of Older Adults along with the HADs scale will be an appropriate next step for future research, followed by a qualitative approach interviewing informal caregivers of AD in order to gain a more in depth understanding.

## Figures and Tables

**Figure 1 medicina-59-00048-f001:**
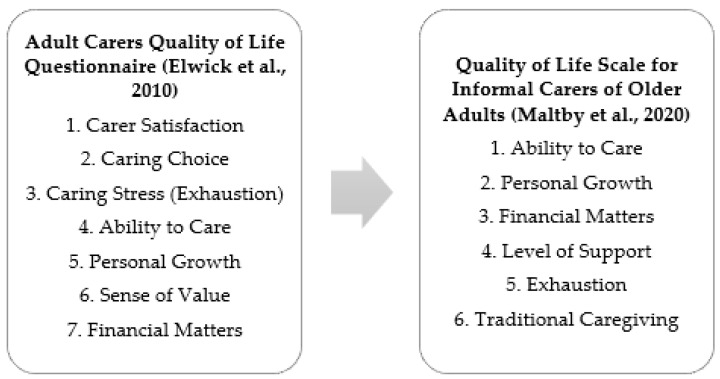
The development of the elements from the Quality of Life Scale for Informal Carers of Older Adults from the Adult Carers Quality of Life Questionnaire [15,16].

**Figure 2 medicina-59-00048-f002:**
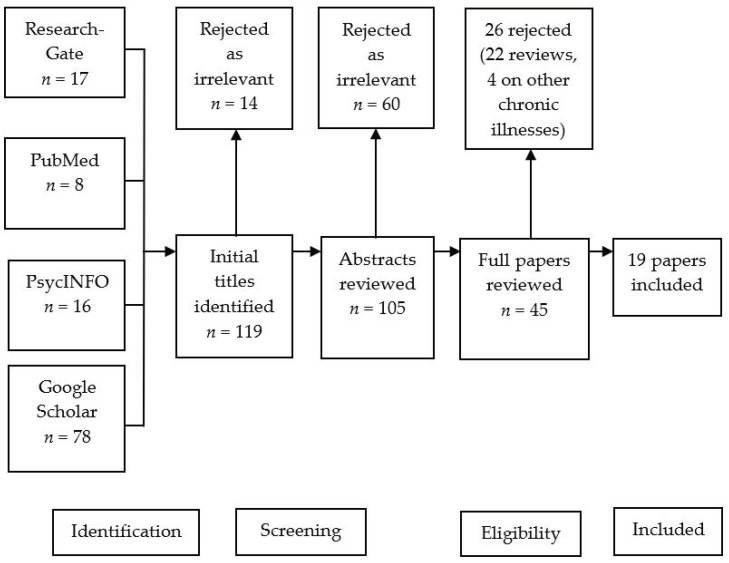
Systematic search process: PRISMA (Preferred Reporting Items for Systematic reviews and Meta-Analyses) flow diagram of article eligibility.

**Table 1 medicina-59-00048-t001:** Search terms.

Participant Identification Terms	Caregiving Terms	Wellbeing Terms	Further Terms
Dementia	Caregiver	Stress	Ability to Care
Alzheimer’s	Carer	Depression	Finance
Alzheimer’s Disease	Informal Caregiver	Anxiety	Money
Elderly	Caregiving	Quality of Life	Personal Growth
Old Age	Support	Depressive Symptoms	Positive Experience
Older Age		Anxious Symptoms	Negative Experience
Cognitive Decline		Mental Health	Traditional Views
		Wellbeing	Non-traditional views
			Caregiving Views
			Ideologies

## Data Availability

Not applicable.

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
