# Peer review of "Informal Caregiving and Alzheimer’s Disease: The Psychological Effect"

_medicina, 2022, doi:10.3390/medicina59010048_

Round 1

Reviewer 1 Report

Overall I liked this paper and would recommend to publish. If you prefer you could change my recommendation to accept with the minor revisions already suggested.   Additionally: 1. How original is the topic? The topic is not original but their purpose was a systematic review. I have not seen a SR on depression.they could reference: Cooper, C.; Balamurali, T.B.S.; Livingston, G. A systematic review of the prevalence and covariates of anxiety in caregivers of people with dementia. International psychogeriatrics 2007, 19, 175-195.   The results should be limited to their findings with comparison with literature in discussion e.g. ref 22 and 28   2. What does it add to the subject area compared with other published material? It adds lots of information using specific QOL scale in relation to anxiety in CGs3. Are the conclusions consistent with the evidence and argumentspresented? yes 4. Do they address the main question posed? The QOL scale addresses 6 areas. They need to highlight exhaustion. In results add to 3.2 and needs report, discussion add to title 4.2

5. line 64 add dressing

6. line 66 add shopping, cooking, cleaning

7. line 289 needs period

Author Response

We thank reviewer for useful comments. We enclose our replies:

Reviewer 1:

Overall I liked this paper and would recommend to publish. If you prefer you could change my recommendation to accept with the minor revisions already suggested.   Additionally: 1. How original is the topic? The topic is not original but their purpose was a systematic review. I have not seen a SR on depression.they could reference:  Cooper, C.; Balamurali, T.B.S.; Livingston, G. A systematic review of the prevalence and covariates of anxiety in caregivers of people with dementia. International psychogeriatrics 200719, 175-195.  

Reply:  We thank the reviewer for this comment. In out study we demonstrate that the caregivers’  experiences, including their mental health in terms of experiencing anxiety and depression, seems to differ depending on their (non-)traditional caregiving ideologies. The methodology between Cooper et al and our study differ, since our systematic was based upon the six elements from the “Quality of Life Scale for Informal Carers of Older Adults” with additional items on traditional and non-traditional caregiving ideologies, as well as caregivers’ experiences. Although it is somewhat difficult to compare the findings between the two studies we feel that our findings are in agreement with the findings of Cooper et al.. we have therefore, referenced this study in the discussion section, and it is now reference number 55.

The results should be limited to their findings with comparison with literature in discussion e.g. ref 22 and 28   

Reply: This has been amended, and now only references 22 and 28 feature.

  1. What does it add to the subject area compared with other published material?It adds lots of information using specific QOL scale in relation to anxiety in CGs
    3. Are the conclusions consistent with the evidence and arguments
    presented? yes
  2. Do they address the main question posed? The QOL scale addresses 6 areas. They need to highlight exhaustion. In results add to 3.2 and needs report, discussion add to title 4.2

Reply:  We have altered the 3.2 and 4.2 subheadings  adding ‘(Exhaustion Factor)’, as suggested by reviewer.

  1. line 64 add dressing

Reply: This has been amended, as recommended.

  1. line 66 add shopping, cooking, cleaning

Reply: This has been amended, as recommended.

  1. line 289 needs period

Reply: This has been amended, as recommended.

Reviewer 2 Report

This review of the literature is limited to published studies on caregiving. No details are provided for the range of publication dates, so there may be earlier publications that were missed. Providing a range of dates for the search would be helpful. It is interesting that many of the studies included focused on Asian or Asian-American caregivers. It implies that more studies are needed that include a broader range of cultural approaches to caregiving. There is no mention of respite care where a volunteer or formal caregiver is assigned for a limited period of time to allow the informal caregiver time away from caring. Such programs have been found to provide needed support and time away from the patient. Also, when speaking of technology, more specific content could be included such as 24/7 caregiver hotlines to provide support when traditional resources are unavailable. Dementia patients often have disturbed sleep, which causes the caregiver to also be up at odd hours. Telephone, computer, or video supports can help caregivers through these difficult times. Further research is needed on the caregivers' view of what supports would be helpful. For example, a solutions-based approach to the Discussion might be helpful. This approach identifies problems but gives possible solutions to address those problems based on the perceived needs of the caregiver.

Author Response

We thank reviewers for useful comments. We enclose our replies:

This review of the literature is limited to published studies on caregiving. No details are provided for the range of publication dates, so there may be earlier publications that were missed. Providing a range of dates for the search would be helpful.

Reply: Thank you for this comment. This is an oversight on our part, not including the information. We have ow attended to this and included the require information in the method section: ‘Systematic literature searches with no limit to study design and published until 01.10.2022’.

It is interesting that many of the studies included focused on Asian or Asian-American caregivers. It implies that more studies are needed that include a broader range of cultural approaches to caregiving. There is no mention of respite care where a volunteer or formal caregiver is assigned for a limited period of time to allow the informal caregiver time away from caring. Such programs have been found to provide needed support and time away from the patient.

Reply: We thank reviewer for this comment. We have now included this in both the discussion section and conclusions.

Also, when speaking of technology, more specific content could be included such as 24/7 caregiver hotlines to provide support when traditional resources are unavailable. Dementia patients often have disturbed sleep, which causes the caregiver to also be up at odd hours. Telephone, computer, or video supports can help caregivers through these difficult times. Further research is needed on the caregivers' view of what supports would be helpful. For example, a solutions-based approach to the Discussion might be helpful. This approach identifies problems but gives possible solutions to address those problems based on the perceived needs of the caregver.

Reply: This is indeed a very helpful comment, and we have taken the liberty to rephrase it at the end of the discussion section.